# Sensor Management with Dynamic Clustering for Bearings-Only Multi-Target Tracking via Swarm Intelligence Optimization

**Xiaoxiao Jiang [1,*], Tianming Ma [1], Jie Jin [1] and Yujie Jiang [2]**

[1] School of Electronic and Electrical Engineering, Shanghai University of Engineering Science, Shanghai 201620, China; tmma@sues.edu.cn (T.M.); jinjie_pku@126.com (J.J.)

[2] School of Computer and Information Engineering, Shanghai Polytechnic University, Shanghai 201209, China; yjjiang@sspu.edu.cn

[*] Correspondence: jxx_simit@163.com; Tel.: +86-21-6779-1084

**Abstract:** Sensor management is a crucial research subject for multi-sensor multi-target tracking in wireless sensor networks (WSNs) with limited resources. Bearings-only tracking produces further challenges related to high nonlinearity and poor observability. Moreover, energy efficiency and energy balancing should be considered for sensor management in WSNs, which involves networking and transmission. This paper formulates the sensor management problem in the partially observable Markov decision process (POMDP) framework and uses the cardinality-balanced multi-target multi-Bernoulli (CBMeMBer) filter for tracking. A threshold control method is presented to reduce the impact on tracking accuracy when using bearings-only measurements for sequential update. Moreover, a Cauchy–Schwarz divergence center is defined to construct a new objective function for efficiently finding the optimal sensor subset via swarm intelligence optimization. This is also conducive to dynamic clustering for the energy efficiency and energy balancing of the network. The simulation results illustrate that the proposed solution can achieve good tracking performance with less energy, and especially that it can effectively balance network energy consumption and prolong network lifetime.

**Keywords:** sensor management; multi-target tracking; random finite set; CBMeMBer; information gain; swarm intelligence optimization

## 1. Introduction

Wireless sensor networks (WSNs) have attracted extensive attention in many fields, such as battlefield surveillance, environmental monitoring, and industrial control [1–6]. Because of rapid deployment, self-organization, low cost, and wide perception range, target tracking in WSNs has more unique application value and prominent superiority [7–10]. Sensor management is very necessary for efficient data fusion in large-scale WSNs due to bandwidth and energy supply constraints. The task of sensor management is to select an optimal subset of sensors to be activated at each time step, then transmit their measurement data to the central processing center, so as to achieve better tracking performance with lower communication consumption. Actually, this is also an equilibrium problem between tracking accuracy and energy consumption.

Multi-target tracking brings great difficulties and challenges to sensor management. Unlike single-target tracking, sensor management for multi-target tracking needs to consider indistinguishable measurement data and resource competition of multiple targets. In general, it comprises two underlying components: multi-target filtering and optimal decision-making [11]. In recent years, multi-target Bayesian filtering based on random finite set (RFS) theory within Mahler's finite set statistics (FISST) framework has been developed [12,13]. As one of the most popular types of FISST, the multi-target

multi-Bernoulli (MeMBer) filter models the multi-target posterior density as multi-Bernoulli RFSs, which can be more accurate and less computational than many other multi-target tracking methods [14]. In order to solve the problem of the cardinality overestimation of the MeMBer filter, the cardinality-balanced MeMBer (CBMeMBer) is proposed as a modified filter and provided sequential Monte Carlo (SMC) and Gaussian mixture (GM) implementations [15]. Following these studies, many advanced filtering methods such as labeled multi-Bernoulli (LMB) filter [16], generalized LMB (GLMB) filter [17], and multi-scan GLMB filter [18] have been developed. The above filters are widely used in various applications for multi-target filtering and good tracking performance.

As mentioned before, decision-making is the other important component of sensor management. Generally, an objective function is developed as the optimization criterion for decision-making. At present, it mainly uses task-based and information-based approaches. A task-based approach is used to formulate the objective function as a cost function, which usually adopts performance metrics such as variance, cardinality estimation, and other distribution-dependent measures [19–24]. Multi-objective optimization can also be considered for sensor management to meet multiple task requirements simultaneously [25–27]. The objective function in an information-based approach is formulated as a reward function that represents the information gain [28], such as Kullback–Leibler (KL) divergence [29–31], Rényi divergence [32–34], and Cauchy–Schwarz (CS) divergence [35–37]. Compared to task-based approaches, information-based approaches can solve or avoid problems such as poor performance for a single task or challenges posed by multiple competing objectives [11,38].

Solving the objective function for sensor management is usually a complicated optimization problem. As the most important type of optimization algorithm, swarm intelligence is inspired by the collective behavior of social insects or animals, which has the advantages of robustness, speed, autonomy, and parallelism. Various representative swarm intelligence optimization algorithms have been put forward so far, such as the particle swarm optimization (PSO) algorithm [39], artificial bee colony (ABC) algorithm [40], social spider optimization (SSO) algorithm [41], firefly algorithm [42], and so on. These algorithms are widely applied to improve objective task accuracy or efficiency in a variety of fields, and they can achieve superior results compared to other optimization methods. Each optimization algorithm provides different performance when dealing with different problems, so an appropriate algorithm needs to be employed depending on the problem and requirement.

In addition, there are two basic types of sensor management problems: single-sensor control and multi-sensor control. For the former, there is only one single mobile sensor in the tracking system, and the sensor state is changed by finding an optimal movement from a set of sensor commands [43,44]. Accordingly, tracking performance and observability can be achieved. In the latter, one or more sensors are selected to observe targets for achieving tracking [45]. Although there are some studies on multi-sensor management for multi-target tracking [46–49], tracking accuracy is still the main focus of algorithm performance, but energy consumption and communication networking are rarely considered. However, besides tracking accuracy, it is also important to take into account the energy efficiency and information transmission of multiple sensors in WSNs, especially network lifetime.

This paper studies sensor management for bearings-only multi-target tracking in WSNs. Bearings-only target tracking has been researched for many years due to its tremendous importance in passive surveillance, which involves estimating the target state only by fusing noisy bearing measurements from multiple different sensors [50]. More importantly, the bearings-only tracking problem is much more complicated and difficult because of the high nonlinearity of angular measurement and poor observability of the target state. Furthermore, the resource consumption for multi-target tracking is commonly unbalanced, because the sensors close to targets will usually be easy to activate and die faster than other sensors due to exhaustion of energy. Meanwhile, sensors will consume excessive energy for communication with the central data processing center. In order to tackle this problem,

the routing schema in two-tier WSNs [51–57] is introduced that divides the activated sensors into cluster member (CM) sensors and cluster head (CH) sensors at each time step. This dynamic clustering strategy can conserve communication bandwidth and balance energy consumption because of short intra-cluster communication and more efficient task allocation, respectively.

Considering the information fusion in WSNs, optimal fusion is unfeasible because of unknown correlations among sensors. In recent years, researchers have shown great interest in suboptimal fusion methods, such as arithmetic mixture densities (AMD) [58,59] and generalized covariance intersection (GCI) [60], which are also called weighted arithmetic average (WAA) [61] and weighted geometric average (WGA) [62]. These methods are designed to combine local multi-sensor estimation results for distributed multi-target tracking, for instance, probability hypothesis densities (PHD) or multi-Bernoulli (MB) densities. On this basis, many fusion mechanisms have been developed and expanded to further improve the estimation performance for various scenarios [63–66]. In this paper, sequential update is used for multi-Bernoulli density fusion by the iterative correction method [11,13]. Although it has no rigorous mathematical derivation, it is easy to implement with low computational complexity and has been widely used in actual applications [11]. However, it is not only sensitive to the update sequence but also easily accumulates estimation errors, especially for bearings-only tracking. This issue will be discussed in a later section.

The key contributions of this paper may be summarized as follows:

- A threshold control method is proposed to reduce the impact of defective bearings-only sensors on multi-target tracking accuracy by SMC-CBMeMBer filtering.
- An information center is defined and formulated to represent an information level in the surveillance area as the foundation of clustering implementation.
- A new objective function is constructed to find the optimal sensor subset around the optimal information center and efficiently solved using the swarm intelligence optimization algorithm.
- The dynamic clustering strategy is utilized to determine CH and CM sensors for balancing energy consumption and prolonging the network lifetime.

The rest of this paper is organized as follows: In Section 2, the necessary background on the multi-target Bayesian framework is introduced, the main steps of the SMC-CBMeMBer filter are described, and the computation of the CS divergence is given. In Section 3, the proposed sensor management solution is presented in detail, including the framework for the optimal sensor subset selection, threshold control for bearings-only measurement, objective function construction, swarm intelligence optimization, dynamic clustering strategy, and the complete solution. Simulations and performance analysis are presented in Section 4. The conclusion is given in Section 5.

## 2. Preliminaries

This section provides the necessary background on the multi-target Bayesian framework using multi-Bernoulli RFS, the SMC-CBMeMBer filter, and CS divergence in order to follow the solution presented throughout this paper.

### 2.1. Multi-Target Bayesian Framework Using Multi-Bernoulli RFS

RFS can be used to construct stochastic models for the multi-target state and multi-target observation due to the uncertainty, such as time-varying target number, missing detections, false alarms, or clutter. In the multi-target system, the posterior multi-target density can be represented as a multi-Bernoulli RFS, which contains not only the multi-target state estimation but also the target number estimation as the cardinality of the RFS.

A multi-Bernoulli RFS is a union of $M$ independent Bernoulli RFSs, and each Bernoulli RFS can be characterized by existence probability $r^{(i)}$ and probability density $p^{(i)}$, $\text{i} = 1, \ldots, M$. Thus, a multi-Bernoulli RFS can be described by using the multi-Bernoulli parameter set $\pi = \left\{ r^{(i)}, p^{(i)} \right\}_{i=1}^{M}$.

Suppose that at time $k$, there are $N(k)$ target states denoted by $x_{k,1}, \ldots, x_{k,N(k)}$ and $M(k)$ measurements denoted by $z_{k,1}, \ldots, z_{k,M(k)}$, hence the multi-target state and the multi-target observation at time $k$ are represented, respectively, by

$$\mathbf{X}_k = \left\{ x_{k,1}, \ldots, x_{k,N(k)} \right\} \in \mathcal{F}(\mathcal{X}) \tag{1}$$

$$\mathbf{Z}_k = \left\{ z_{k,1}, \ldots, z_{k,M(k)} \right\} \in \mathcal{F}(\mathcal{Z}) \tag{2}$$

where $\mathcal{X} \subseteq \mathbb{R}^{n_x}$ is the state space, $\mathcal{Z} \subseteq \mathbb{R}^{n_z}$ is the observation space, $\mathcal{F}(\mathcal{X})$ and $\mathcal{F}(\mathcal{Z})$ are the finite subsets of $\mathcal{X}$ and $\mathcal{Z}$, respectively.

The multi-target filtering problem can be posed in the Bayesian framework. If the posterior multi-target density at time $k$ is denoted by $\pi_k(\mathbf{X}_k|\mathbf{Z}_{1:k})$, then the multi-target Bayes recursion propagates $\pi_k(\mathbf{X}_k|\mathbf{Z}_{1:k})$ containing two main steps, prediction and update, which are as follows:

$$\pi_{k-1|k}(\mathbf{X}_k|\mathbf{Z}_{1:k-1}) = \int f_{k|k-1}(\mathbf{X}_k|\mathbf{X}) \pi_{k-1}(\mathbf{X}|\mathbf{Z}_{1:k-1}) \delta\mathbf{X} \tag{3}$$

$$\pi_k(\mathbf{X}_k|\mathbf{Z}_{1:k}) = \frac{g_k(\mathbf{Z}_k|\mathbf{X}_k) \pi_{k-1|k}(\mathbf{X}_k|\mathbf{Z}_{1:k-1})}{\int g_k(\mathbf{Z}_k|\mathbf{X}) \pi_{k-1|k}(\mathbf{X}|\mathbf{Z}_{1:k-1}) \delta\mathbf{X}} \tag{4}$$

where $f_{k|k-1}(\cdot|\cdot)$ is the multi-target transition density and $g_k(\cdot|\cdot)$ is the multi-target likelihood. The multi-target transition implies the model of target motions, births, and deaths. The multi-target likelihood involves the multi-target observation model, which is formed by the union of target detections and false alarms or clutter. The expressions for $f_{k|k-1}(\mathbf{X}_k|\mathbf{X}_{k-1})$ and $g_k(\mathbf{Z}_k|\mathbf{X}_k)$ can be derived from the multi-target system model using FISST. In addition, the integrals in the above recursion are FISST integrals.

### 2.2. SMC-CBMeMBer Filter

As we know, the above Bayes recursion has no closed-form analytic solution in general. In this paper, we consider using the SMC-CBMeMBer filter to achieve multi-target tracking, which is the most popular implementation of the multi-target Bayesian filter. For the sake of a clear description in the later section, the main steps and parameters of the SMC-CBMeMBer filter are given below (for more details, see [15]).

(1)  Prediction:

Suppose that at time $k-1$ the posterior multi-Bernoulli multi-target density is given by $\pi_{k-1} = \left\{ (r_{k-1}^{(i)}, p_{k-1}^{(i)}) \right\}_{i=1}^{M_{k-1}}$, where $M_{k-1}$ is the number of Bernoulli RFSs, each $r_{k-1}^{(i)}$ and $p_{k-1}^{(i)}$ are existence probability and probability density of the $i_{th}$ Bernoulli RFS, respectively, and $p_{k-1}^{(i)}$ is comprised of a set of weighted particle samples $\left\{ w_{k-1}^{(i,j)}, x_{k-1}^{(i,j)} \right\}_{j=1}^{L_{k-1}^{(i)}}$, which can be written as

$$p_{k-1}^{(i)} = \sum_{j=1}^{L_{k-1}^{(i)}} w_{k-1}^{(i,j)} \delta_{x_{k-1}^{(i,j)}}(x) \tag{5}$$

where $L_{k-1}^{(i)}$ is the number of particle samples, and $w_{k-1}^{(i,j)}$ is the weight of the $j_{th}$ particle corresponding to the $i_{th}$ Bernoulli RFS. Then, the predicted multi-target density is also a multi-Bernoulli form, written as

$$\pi_{k|k-1} = \left\{ (r_{P,k|k-1}^{(i)}, p_{P,k|k-1}^{(i)}) \right\}_{i=1}^{M_{k-1}} \cup \left\{ (r_{\tau,k}^{(i)}, p_{\tau,k}^{(i)}) \right\}_{i=1}^{M_{\tau,k}} \tag{6}$$

where $\left\{ \left( r_{P,k|k-1}^{(i)}, p_{P,k|k-1}^{(i)} \right) \right\}_{i=1}^{M_{k-1}}$ and $\left\{ \left( r_{\tau,k}^{(i)}, p_{\tau,k}^{(i)} \right) \right\}_{i=1}^{M_{\tau,k}}$ are the multi-Bernoulli parameter sets for the surviving targets and target births, respectively. Given the importance density $q_k^{(i)}\left( \cdot \middle| x_{k-1}, Z_k \right)$ and $b_k^{(i)}(\cdot | Z_k)$, then $\pi_{k|k-1}$ can be computed as follows:

$$r_{P,k|k-1}^{(i)} = r_{k-1}^{(i)} \sum\nolimits_{j=1}^{L_{k-1}^{(i)}} w_{k-1}^{(i,j)} p_{s,k}(x_{k-1}^{(i,j)}) \tag{7}$$

$$p_{P,k|k-1}^{(i)}(x) = \sum\nolimits_{j=1}^{L_{k-1}^{(i)}} \widetilde{w}_{P,k|k-1}^{(i,j)} \delta_{x_{P,k|k-1}^{(i,j)}}(x) p_{s,k}(x_{k-1}^{(i,j)}) \tag{8}$$

$$r_{\tau,k}^{(i)} = parameter\ given\ by\ birth\ model \tag{9}$$

$$p_{\tau,k}^{(i)}(x) = \sum\nolimits_{j=1}^{L_{\tau,k}^{(i)}} \widetilde{w}_{\tau,k}^{(i,j)} \delta_{x_{\tau,k}^{(i,j)}}(x) \tag{10}$$

where $p_{s,k}(x)$ is the target survival probability at time $k$ given previous state $x$, $x_{P,k|k-1}^{(i,j)}$ and $x_{\tau,k}^{(i,j)}$ are the surviving target state and the birth target state, which are sampled according to $q_k^{(i)}\left( \cdot \middle| x_{k-1}^{(i,j)}, Z_k \right)$ and $b_k^{(i)}(\cdot | Z_k)$, respectively,

$$w_{P,k|k-1}^{(i,j)} = \frac{w_{k-1}^{(i,j)} f_{k|k-1}(x_{P,k|k-1}^{(i,j)} \middle| x_{k-1}^{(i,j)}) p_{s,k}(x_{k-1}^{(i,j)})}{q_k^{(i)}(x_{P,k|k-1}^{(i,j)} \middle| x_{k-1}^{(i,j)}, Z_k)} \tag{11}$$

$$\widetilde{w}_{P,k|k-1}^{(i,j)} = w_{P,k|k-1}^{(i,j)} / \sum\nolimits_{j=1}^{L_{k-1}^{(i)}} w_{P,k|k-1}^{(i,j)} \tag{12}$$

$$w_{\tau,k}^{(i,j)} = \frac{p_{\tau,k}^{(i)}(x_{\tau,k}^{(i,j)})}{b_k^{(i)}(x_{\tau,k}^{(i,j)} \middle| Z_k)} \tag{13}$$

$$\widetilde{w}_{\tau,k}^{(i,j)} = w_{\tau,k}^{(i,j)} / \sum\nolimits_{j=1}^{L_{\tau,k}^{(i)}} w_{\tau,k}^{(i,j)} \tag{14}$$

(2)　Update:

Suppose that at time $k$, the predicted multi-target density is denoted as $\pi_{k|k-1} = \left\{ \left( r_{k|k-1}^{(i)}, p_{k|k-1}^{(i)} \right) \right\}_{i=1}^{M_{k|k-1}}$ and each $p_{k|k-1}^{(i)}$ is comprised of a set of weighted particle samples $\left\{ w_{k|k-1}^{(i,j)}, x_{k|k-1}^{(i,j)} \right\}_{j=1}^{L_{k|k-1}^{(i)}}$, which can be given by

$$p_{k|k-1}^{(i)} = \sum\nolimits_{j=1}^{L_{k|k-1}^{(i)}} w_{k|k-1}^{(i,j)} \delta_{x_{k|k-1}^{(i,j)}}(x) \tag{15}$$

Then, the posterior multi-Bernoulli multi-target density can be denoted by

$$\pi_k = \left\{ \left( r_{L,k}^{(i)}, p_{L,k}^{(i)} \right) \right\}_{i=1}^{M_{k|k-1}} \cup \left\{ \left( r_{U,k}(z), p_{U,k}(\cdot;z) \right) \right\}_{z \in Z_k} \tag{16}$$

which is formed by the union of the multi-Bernoulli parameter sets for the legacy tracks $\left( \left\{ \left( r_{L,k}^{(i)}, p_{L,k}^{(i)} \right) \right\}_{i=1}^{M_{k|k-1}} \right)$ and measurement-updated tracks $\left( \left\{ \left( r_{U,k}(z), p_{U,k}(\cdot;z) \right) \right\}_{z \in Z_k} \right)$, where

$$r_{L,k}^{(i)} = r_{k|k-1}^{(i)} \frac{1 - \rho_{L,k}^{(i)}}{1 - r_{k|k-1}^{(i)} \rho_{L,k}^{(i)}} \tag{17}$$

$$p_{L,k}^{(i)}(x) = \sum_{j=1}^{L_{k|k-1}^{(i)}} \widetilde{w}_{L,k}^{(i,j)} \delta_{x_{k|k-1}^{(i,j)}}(x) \tag{18}$$

$$r_{U,k}(z) = \frac{\sum_{i=1}^{M_{k|k-1}} \frac{r_{k|k-1}^{(i)}(1-r_{k|k-1}^{(i)})\rho_{U,k}^{(i)}(z)}{\left(1-r_{k|k-1}^{(i)}\rho_{L,k}^{(i)}\right)^2}}{\kappa_k(z) + \sum_{i=1}^{M_{k|k-1}} \frac{r_{k|k-1}^{(i)}\rho_{U,k}^{(i)}(z)}{1-r_{k|k-1}^{(i)}\rho_{L,k}^{(i)}}} \tag{19}$$

$$p_{U,k}(x;z) = \sum_{i=1}^{M_{k|k-1}} \sum_{j=1}^{L_{k|k-1}^{(i)}} \widetilde{w}_{U,k}^{*(i,j)}(z) \delta_{x_{k|k-1}^{(i,j)}}(x) \tag{20}$$

where

$$\rho_{L,k}^{(i)} = \sum_{j=1}^{L_{k|k-1}^{(i)}} w_{k|k-1}^{(i,j)} p_{D,k}(x_{k|k-1}^{(i,j)}) \tag{21}$$

$$\widetilde{w}_{L,k}^{(i,j)} = w_{L,k}^{(i,j)} / \sum_{j=1}^{L_{k|k-1}^{(i)}} w_{L,k}^{(i,j)} \tag{22}$$

$$w_{L,k}^{(i,j)} = w_{k|k-1}^{(i,j)}(1 - p_{D,k}(x_{k|k-1}^{(i,j)})) \tag{23}$$

$$\rho_{U,k}^{(i)}(z) = \sum_{j=1}^{L_{k|k-1}^{(i)}} w_{k|k-1}^{(i,j)} \varphi_{k,z}(x_{k|k-1}^{(i,j)}) \tag{24}$$

$$\widetilde{w}_{U,k}^{*(i,j)}(z) = w_{U,k}^{*(i,j)}(z) / \sum_{i=1}^{M_{k|k-1}} \sum_{j=1}^{L_{k|k-1}^{(i)}} w_{U,k}^{*(i,j)}(z) \tag{25}$$

$$w_{U,k}^{*(i,j)}(z) = w_{k|k-1}^{(i,j)} \frac{r_{k|k-1}^{(i)}}{1-r_{k|k-1}^{(i)}} \varphi_{k,z}(x_{k|k-1}^{(i,j)}) \tag{26}$$

$$\varphi_{k,z}(x_{k|k-1}^{(i,j)}) = g_k(z|x_{k|k-1}^{(i,j)}) \, p_{D,k}(x_{k|k-1}^{(i,j)}) \tag{27}$$

where $p_{D,k}(x)$ and $g_k(\cdot|x)$ are the target detection probability and the single target measurement likelihood, respectively, at time $k$ given current state $x$, and $\kappa_k(z)$ is the intensity of Poisson clutter at time $k$.

In order to reduce the degeneracy of the particle samples, a resampling step is used for each hypothesized track after the update. Pruning is also performed to eliminate the hypothesized tracks with low existence probabilities. More details are given in [15].

### 2.3. CS Divergence

CS divergence is commonly used as an information divergence measure because it is more amenable to a closed-form analytical solution compared with Kullback–Leibler divergence or Rényi divergence [38]. This paper adopts the CS divergence as the information gain metric for sensor management. Here, the computation of the CS divergence is directly given below [37].

In order to more clearly formulate the CS divergence, some variables explained before are described again. As mentioned earlier, the predicted multi-Bernoulli multi-target density is $\pi_{k|k-1} = \left\{ (r_{k|k-1}^{(i)}, p_{k|k-1}^{(i)}) \right\}_{i=1}^{M_{k|k-1}}$, and each $p_{k|k-1}^{(i)}$ is comprised of a set of weighted samples $\left\{ w_{k|k-1}^{(i,j)}, x_{k|k-1}^{(i,j)} \right\}_{j=1}^{L_{k|k-1}^{(i)}}$. The updated posterior multi-Bernoulli multi-target density is

$\pi_k = \left\{ \left( r_{L,k}^{(i)}, p_{L,k}^{(i)} \right) \right\}_{i=1}^{M_{k|k-1}} \cup \left\{ \left( r_{U,k}(z), p_{U,k}(\cdot; z) \right) \right\}_{z \in \mathbf{Z}_k}$, where each $p_{L,k}^{(i)}$ and $p_{U,k}(\cdot; z)$ can be

approximated by a set of weighted samples $\left\{ \widetilde{w}_{L,k}^{(i,j)}, x_{k|k-1}^{(i,j)} \right\}_{j=1}^{L_{k|k-1}^{(i)}}$ and $\left\{ \widetilde{w}_{U,k}^{*(i,j)}(z), x_{k|k-1}^{(i,j)} \right\}_{j=1}^{L_{k|k-1}^{(i)}}$.

Then, the CS divergence between $\pi_{k|k-1}$ and $\pi_k$ can be given by

$$D_{CS}\left( \pi_{k|k-1}, \pi_k \right) = 1/2 \times K \times \sum_{i=1}^{M_{k|k-1}} \sum_{j=1}^{L_{k|k-1}^{(i)}} \left[ r_{L,k}^{(i)} \widetilde{w}_{L,k}^{(i,j)} + \sum_{z \in \mathbf{Z}_k} r_{U,k}^{(i)}(z) \widetilde{w}_{U,k}^{*(i,j)}(z) - r_{k|k-1}^{(i)} w_{k|k-1}^{(i,j)} \right]^2 \quad (28)$$

where *K* is the unit of (hyper-volume) measurement in a single-target state. Here, note that the particles do not change through the update in the SMC-CBMeMBer filter, but the number of the Bernoulli components changes.

## 3. Sensor Management Solution

In this section, we briefly explain the framework of sensor management for bearings-only multi-target tracking and discuss the issues to be addressed in the solution design. Then, a threshold control method is proposed as an improvement of bearings-only sensor selection. Subsequently, the objective function construction and optimization are given to achieve sensor selection. After that, a dynamic clustering strategy is introduced for data transmission, and then multi-sensor multi-Bernoulli density fusion is given to implement SMC-CBMeMBer filtering. Finally, the complete proposed sensor management solution is given.

### 3.1. Sensor Management for Bearings-Only Multi-Target Tracking

It is important to note that just one sensor's measurement data are used to complete the update steps in the general SMC-CBMeMBer filter. However, one sensor is not enough for bearings-only multi-target tracking, which cannot obtain the only determination solution of target state estimation without range information due to state non-observability. Therefore, we use sequential update with multiple sensors to achieve tracking. The details of the sequential update will be given later.

This paper aims to design an energy-efficient sensor management solution that can provide good tracking performance by selecting an optimal subset of bearings-only sensors to be activated. It is formulated in the partially observable Markov decision process (POMDP) framework [67]. In the POMDP framework, the sensor selection is perceived from the previous measurements. Hence, the POMDP of sensor selection in this paper can be defined as

$$\Psi = \left\{ \mathbf{X}_k, \mathbb{S}_k^C, f_{k|k-1}(x_k|x_{k-1}), \mathbf{Z}, h_k(z|x; s), \vartheta(\mathbb{S}_k^A, \mathbf{X}_k) \right\} \quad (29)$$

where $\mathbb{S}_k^C$ is a finite set of candidate sensors at time *k*, $f_{k|k-1}(x_k|x_{k-1})$ is a transition model for the single-target state, $\mathbf{Z}$ is a finite set of measurements, $\vartheta(\mathbb{S}_k^A, \mathbf{X}_k)$ is an objective function that computes a reward or a cost of selecting a sensor subset $\mathbb{S}_k^A, \mathbb{S}_k^A \subseteq \mathbb{S}_k^C$, and $h_k(z|x; s)$ is an observation model with sensor $s \in \mathbb{S}_k^A$.

The essential part of sensor selection is usually modeled as the optimization of an objective function. Accordingly, the purpose of the optimization is to find the optimal sensor subset $\mathbb{S}_k^{A^*}$ to maximize or minimize the statistical expectation of the objective function $\vartheta(\mathbb{S}_k^A, \mathbf{X}_k)$ over all measurements:

$$\mathbb{S}_k^{A^*} = \underset{\mathbb{S}_k^A \subseteq \mathbb{S}_k^C}{\arg\max} / \underset{\mathbb{S}_k^A \subseteq \mathbb{S}_k^C}{\arg\min} \left\{ \mathbb{E}_z[\vartheta(\mathbb{S}_k^A, \mathbf{X}_k)] \right\} \quad (30)$$

When the sensor subset $\mathbb{S}_k^A$ is substituted in the objective function for optimization, its corresponding measurements are unknown. Therefore, a pseudo-measurement set is used instead of unknown future measurements. To reduce computational complexity, the predicted ideal measurement set (PIMS) is generated by the observation model; note that it is clutter-free and noise-free. The diagram of sensor management within the POMDP framework in this paper is shown in Figure 1, in which the number of candidate sensors is denoted as $N_c$.

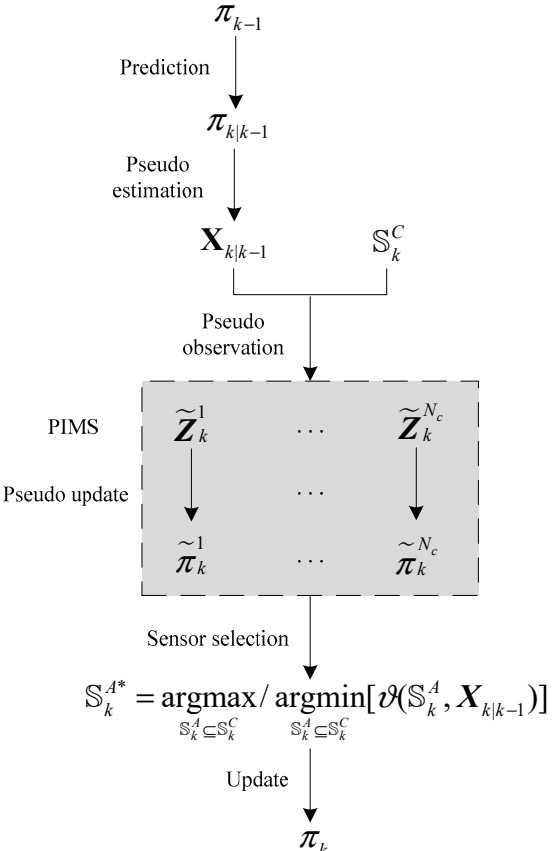

**Figure 1.** Sensor management for bearing-only multi-target tracking within the POMDP framework.

Generally, the sensors with high information gain, such as CS divergence, can be selected for possibly reachable tracking accuracy. However, it is not reasonable to only consider tracking accuracy for sensor selection, especially for bearings-only multi-target tracking in WSNs. There are two issues that need to be addressed in this paper, as follows:

- Although multiple sensors are selected to meet the state observability, the quality of reserved particle samples within the sequential update process will still be impacted because only one sensor's measurements can be used at a time. This might result in reduced tracking accuracy and even loss of targets.
- The selected sensors will transmit their measurement data and consume enormous energy. Meanwhile, the energy consumption is usually unbalanced due to the target motion path and sensor deployment. This will speed up energy exhaustion and shorten the network lifetime.

### 3.2. Objective Function and Swarm Intelligence Optimization

#### 3.2.1. Threshold Control for Bearings-Only Measurement

There is an intractable problem for bearings-only tracking using the SMC-CBMeMBer filter, shown in Figure 2. In Figure 2, the sensor's measurement data are the angles between the direction of the target and the positive y-axis in a two-dimensional system. It can be observed that the measurement difference between different target positions is very small for Sensor 2. This will easily result in the failure of the target state estimate because similar measurements cannot distinguish different target positions. When Sensor 2 is selected to calculate the measurement likelihood for the update of SMC-CBMeMBer filtering, some wrong particle samples will have large weight and be reserved after resampling. By comparison, it is significant that Sensor 1 is better than Sensor 2 for tracking accuracy. However, it is quite possible that Sensor 2 will be selected because of its high CS divergence. Therefore, it is necessary to reduce the impact of such sensors for sensor selection.

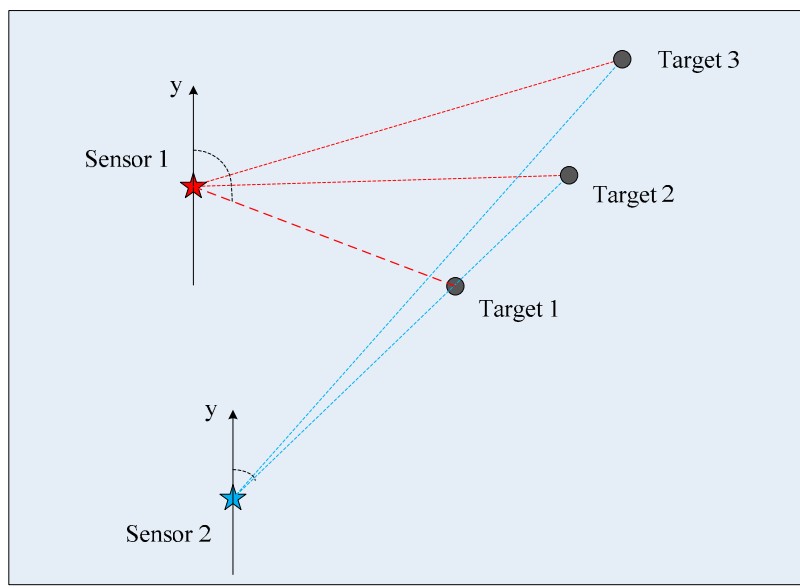

**Figure 2.** The problem of bearings-only measurements using the SMC-CBMeMBer filter. Stars with different colors are used to represent sensors at different positions.

As mentioned before, the CS divergence of each candidate sensor $s$ is used for sensor selection, which can be computed by

$$D_{CS}^s = 1/2 \times K \times \sum_{i=1}^{M_{k|k-1}} \sum_{j=1}^{L_{k|k-1}^{(i)}} \left[ r_{L,k}^{(i)} \widetilde{w}_{L,k}^{(i,j)} + \sum_{z \in \widetilde{Z}_k^s} r_{U,k}^{(i)}(z) \widetilde{w}_{U,k}^{*(i,j)}(z) - r_{k|k-1}^{(i)} w_{k|k-1}^{(i,j)} \right]^2 \tag{31}$$

where $\widetilde{Z}_k^s$ is the PIMS of sensor $s$ at time $k$, which can be obtained by the observation model with the predicted multi-target state $X_{k|k-1}$ obtained from $\pi_{k|k-1}$.

Here, two control variables are designed to tackle the above bearings-only problem. One is the predicted ideal measurement difference threshold $\varepsilon_z$, and the other is the predicted ideal target position difference threshold $\varepsilon_x$. The proposed threshold control method is as follows:

Compute the predicted ideal target state set $PITS_k$ using the state transition model with previous posterior state estimation $X_{k-1|k-1}$. The number of target states in $PITS_k$ is denoted as $N_{PITS}^k$, then the predicted target position difference can be computed by

$$x_{diff}^k(i,j) = \sqrt{\left(PITS_k^i(1) - PITS_k^j(1)\right)^2 + \left(PITS_k^i(2) - PITS_k^j(2)\right)^2}, \; i \neq j \tag{32}$$

where $i = 1, \ldots, N_{PITS}^k$, $j = 1, \ldots, N_{PITS}^k$, $\left[PITS_k^i(1), PITS_k^i(2)\right]$ and $\left[PITS_k^j(1), PITS_k^j(2)\right]$ are the two-dimensional positions of the $i_{th}$ and the $j_{th}$ predicted target states at time $k$, respectively.

For each candidate sensor $s \in \mathbb{S}_k^C$, the predicted ideal measurement set $PIMS_k^s$ can be computed by the observation model with $PITS_k$, and then the predicted ideal measurement difference can be obtained by

$$z_{diff}^{s,k}(i,j) = \left| PIMS_k^s(i) - PIMS_k^s(j) \right|, \; i \neq j \tag{33}$$

where $PIMS_k^s(i)$ and $PIMS_k^s(j)$ correspond to $PITS_k^i$ and $PITS_k^j$, respectively.

For any $(i,j)$ with both $z_{diff}^{s,k} < \varepsilon_z$ and $x_{diff}^k > \varepsilon_x$ being satisfied simultaneously, the CS divergence of sensor $s$ denoted by $D_{CS}^s$ should be decreased to reduce its possibility of being selected. The following formula is used to implement the threshold control:

$$D_{CS}^s = w * D_{CS}^s \tag{34}$$

where $w \in [0,1]$, and it is better to define it as a very small number. To simplify, $w$ is set to 0 in this paper.

### 3.2.2. Objective Function Construction

In order to find the optimal sensor subset to meet the requirements of tracking accuracy and communication consumption, the objective function of sensor selection needs to have a more specific definition.

In the surveillance area, each position in the coordinate system is defined as an information center, which can be also called a CS divergence center (CSC) because the CS divergence is used as the information metric in this paper. Each CSC has a corresponding CS divergence level formulated as follows:

$$levelD_{CS}^c = sum(D_{CS}^s), s \in \mathbb{S}_k^A(c) \tag{35}$$

where $c$ represents the position of the CSC, $\mathbb{S}_k^A(c)$ is the sensor set containing the specified number of candidate sensors around the position $c$, $s$ is the sensor that belongs to $\mathbb{S}_k^A(c)$, $sum(D_{CS}^s)$ represents the sum of the CS divergences of all sensors in $\mathbb{S}_k^A(c)$.

Assume that there are $N_a$ sensors that need to be selected and activated. Then, the sensor set $\mathbb{S}_k^A(c)$ is formed by the sensors with the shortest $N_a$ distance from position $c$. Suppose the initial $\mathbb{S}_k^A(c)$ is empty, then the sensor set $\mathbb{S}_k^A(c)$ can be obtained by

$$repeat\ N_a\ times \begin{cases} s = \text{argmin}_{s \in \{\mathbb{S}_k^C \setminus \mathbb{S}_k^A(c)\}} dis_c^s \\ \mathbb{S}_k^A(c) = \mathbb{S}_k^A(c) \cup \{s\} \end{cases} \tag{36}$$

$$dis_c^s = \sqrt{(c_1 - x^s)^2 + (c_2 - y^s)^2} \tag{37}$$

where $[c_1, c_2]$ represents the position $c$ in a two-dimensional coordinate system, $c_1$ and $c_2$ are the positions of the x-axis and y-axis, respectively, and similarly $[x^s, y^s]$ are the coordinates of the position of the sensor $s$.

Then, we can define the objective function to be the CS divergence level $levelD_{CS}^c$ as

$$f(c) = levelD_{CS}^c \tag{38}$$

The optimization of the objective function is to find the maximum $levelD_{CS}^c$ with the optimal position $c^*$, which can be described by

$$c^* = argmax\ f(c) = argmax\ levelD_{CS}^c \tag{39}$$

Now, the problem becomes how to efficiently find the optimal solution for this objective function.

### 3.2.3. Optimization

As one of the most common swarm intelligence algorithms, the PSO algorithm has the advantages of easy implementation, high-quality solutions, computational efficiency, and convergence speed to solve various optimization problems [68]. The details of the general PSO algorithm will not be described again here, but it is important to make clear how to implement the PSO algorithm for solving the optimization problem in this paper, which is briefly described as follows:

Step 0: Set the initial iteration $l = 0$. Randomly generate the initial population $pop = \left[X_1^0, X_2^0, \ldots, X_{popsize}^0\right]^T$, $popsize$ is the population size, which is also the number of particles. Each particle $X_i^0 = \left[c_{i,1}^0, c_{i,2}^0, \ldots, c_{i,D}^0\right]$, $i = 1, 2, \ldots, popsize$ represents its position vector in the surveillance area, and $D = 2$ in the two-dimensional system.

Note that the particle in PSO is different from that in SMC-CBMeMBer. They are distinguished by their meanings and representations. The former is an individual of the population in the swarm intelligence optimization, the latter is a sample of a target state for multi-target filtering.

Here, each particle in PSO has its initial velocity $v_{i,j}^0 = 0$, $i = 1, 2, \ldots, popsize$, $j = 1, 2 \ldots, D$. Then, set each particle's history best position $pbest_i^0 = X_i^0$, and set global best position $gbest = argmax f(X_i^0) = argmax levelD_{CS}^{X_i^0}$.

Step 1: Update the iteration time $l = l + 1$.

Step 2: For each particle, update its velocity and position:

$$v_{i,j}^l = \omega \cdot v_{i,j}^{l-1} + \varphi_1 r_1 (pbest_i^{l-1} - X_i^{l-1}) + \varphi_2 r_2 (gbest - X_i^{l-1}) \tag{40}$$

$$c_{i,j}^l = c_{i,j}^{l-1} + v_{i,j}^l \tag{41}$$

where $\omega$ is the inertia weight, $\varphi_1$ and $\varphi_2$ are the acceleration coefficients, $r_1$ and $r_2$ are two random numbers uniformly distributed in [0, 1].

Step 3: Evaluate the updated population in terms of the fitness value, which can be written by the objective function as

$$f(X_i^l) = levelD_{CS}^{X_i^l} \tag{42}$$

Then, we can use Equations (35)–(37) to calculate the value of this function.

Step 4: Update each particle's history best position $pbest_i^l$. For each particle, if the $f(X_i^l) > pbest_i^l$, then $pbest_i^l = X_i^l$.

Step 5: Update the global best position $gbest$ as $gbest = \mathrm{argmax} f(pbest_i^l)$.

Step 6: If the convergence criteria are satisfied, then terminate. Otherwise, go to Step 1.

After the above steps, we can obtain the optimal position $c^* = gbest$, and the optimal sensor subset $\mathbb{S}_k^{A^*}$.

### 3.3. Dynamic Clustering Strategy

The constructed objective function considers not only tracking accuracy by high CS divergence level but also energy efficiency by clustering with the short-distance intra-cluster communication. After the optimal sensor subset is determined, the selected sensors will be activated by the base station (BS), and then they can observe multiple targets in the surveillance area and transmit their measurement data. In order to save energy consumption and prolong the network lifetime, we adopt a dynamic clustering strategy to complete the measurement data transmission. Figure 3 shows the cluster structure and workflow used in this paper.

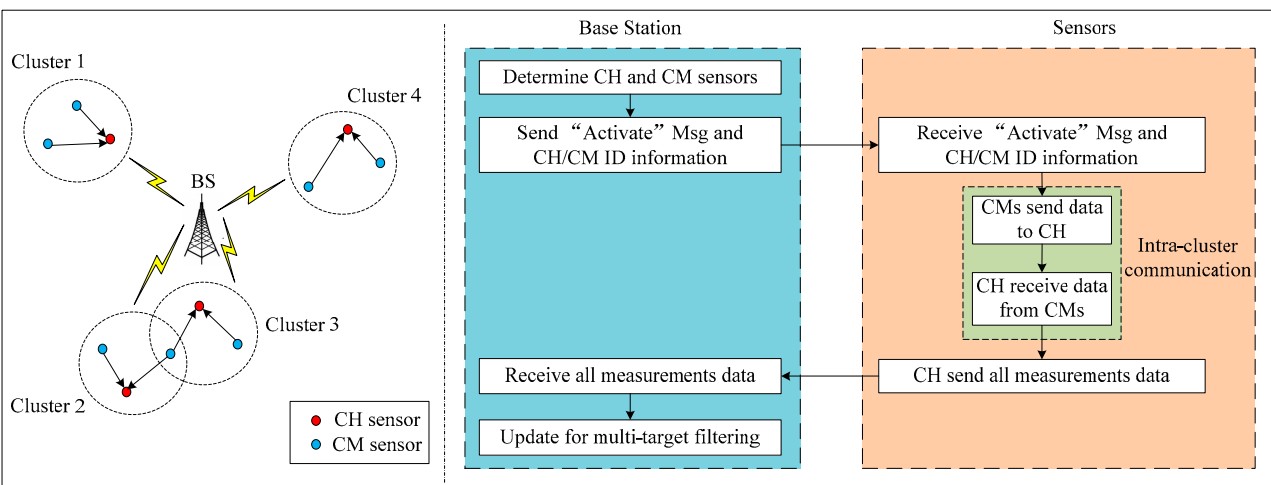

**Figure 3.** Cluster structure and workflow.

At each time step, the activated sensors dynamically form a new cluster. The sensor with the most remaining energy is the CH sensor, and the other sensors are CM sensors. The CM sensors send their measurement data to the CH sensor, and then the CH sensor sends all data containing its own and received data to the base station. This dynamic clustering strategy will benefit energy efficiency and energy balance for two reasons. On the one hand, the selected sensors are close in distance according to the above objective function model, so the intra-cluster communication will consume less energy with the short distance. On the other hand, the CH sensor with the most remaining energy will consume more energy due to its transmission to the base station, and the CM sensors can save energy for balancing network energy consumption.

The complete dynamic clustering strategy is described as follows:

Step 1: The base station determines the CH and CM sensors by the remaining energy of each sensor in $\mathbb{S}_k^{A^*}$, and then sends a message "Activate" and the CH/CM ID information to the selected sensors.

Step 2: The selected sensors receive the message "Activate" and the CH/CM ID information, and then a new cluster is formed.

Step 3: All activated sensors sense targets and obtain their measurements with their own data processing units. The CM sensors send their measurements to the CH sensor, then the CH sensor sends all measurement data to the base station.

Step 4: The base station runs the SMC-CBMeMBer filter using received measurement data to obtain the multi-target state estimation. Then, it returns to sensor selection optimization for tracking at the next time point.

### 3.4. Multi-Sensor Multi-Bernoulli Density Fusion with Sequential Update

The SMC-CBMeMBer filter uses sequential update for multi-sensor multi-Bernoulli density fusion by the iterative correction method [13]. Using multiple sensors has the advantages of flexibility, robustness, and fault tolerance, especially because it can meet the requirements of the state observability for bearings-only tracking.

Assume that there are $N_a$ sensors activated to participate in tracking, the measurement of sensor $s$ is denoted by $\boldsymbol{Z}_k^s = \{z_{k,1}^s, \ldots, z_{k,M_k^s}^s\}$, where $M_k^s$ is the number of measurements; hence, the measurement set of all active sensors is denoted by $\mathbb{Z}_k = \left\{ \boldsymbol{Z}_k^1, \ldots, \boldsymbol{Z}_k^s, \ldots, \boldsymbol{Z}_k^{N_a} \right\}$.

The process of the multi-sensor multi-Bernoulli density fusion with sequential update is illustrated in Figure 4. For simplicity, the number of active sensors here is set to $N_a = 3$. In the process of multi-target density update by $\mathbb{Z}_k$, only one sensor's measurements are used at a time. After one update, the updated multi-target density can be taken as the predicted multi-target density for the next update. Finally, the multi-Bernoulli multi-target density $\pi_{k|k}$ can be obtained by sequential update.

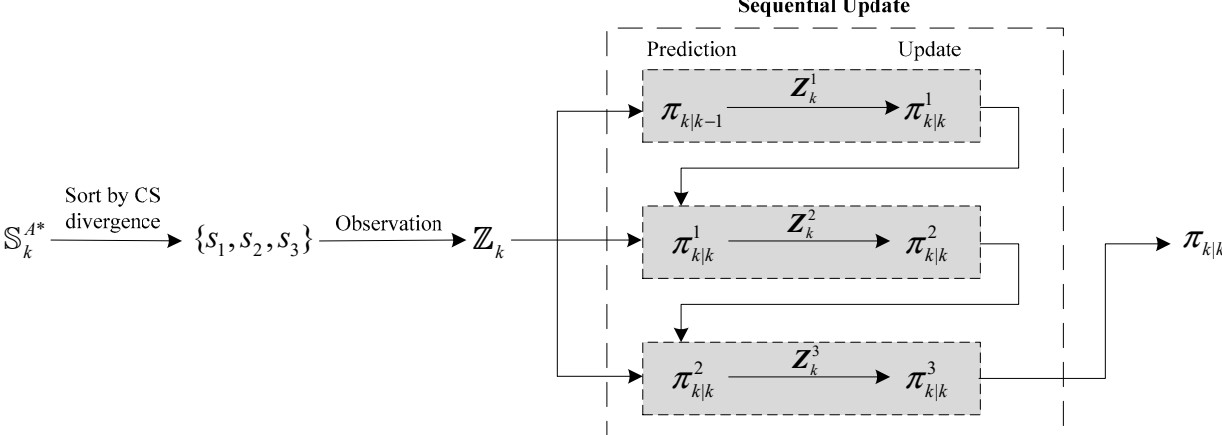

**Figure 4.** The process of the multi-sensor multi-Bernoulli density fusion with sequential update.

In addition, the update sequence is very important for tracking performance [13]. Therefore, in this paper, the measurement set $\mathbb{Z}_k$ is ranked according to the CS divergence of the selected sensors. The measurement of the sensor with the minimum CS divergence is used first for the update, and the one with the maximum CS divergence is updated last. This can maximize the potential tracking accuracy using CS divergence.

### 3.5. The Complete Solution

The complete sensor management solution proposed in this paper is as follows (Algorithm 1):

---

**Algorithm 1:** The complete sensor management solution

---

**Input:** multi-target density $\pi_{k-1} = \left\{ (r_{k-1}^{(i)}, p_{k-1}^{(i)}) \right\}_{i=1}^{M_{k-1}}$, multi-target state estimate $X_{k-1|k-1}$, candidate sensor set $\mathbb{S}_k^C$, and the remaining energy of each sensor $s \in \mathbb{S}_k^C$.
**Solution:**
1: Execute the prediction steps of SMC-CBMeMBer, then obtain the predicted multi-target density $\pi_{k|k-1} = \left\{ (r_{k|k-1}^{(i)}, p_{k|k-1}^{(i)}) \right\}_{i=1}^{M_{k|k-1}}$;
2: For each Bernoulli RFS $i$, compute $X_{k|k-1}^{(i)}$ by $p_{k|k-1}^{(i)}$, which is comprised of weighted samples $\left\{ w_{k|k-1}^{(i,j)}, x_{k|k-1}^{(i,j)} \right\}_{j=1}^{L_{k|k-1}^{(i)}}$. For each sensor $s \in \mathbb{S}_k^C$, compute its corresponding PIMS $\widetilde{Z}_k^s$ by the observation model with $X_{k|k-1}$. Then, update $\pi_{k|k-1}$ by $\widetilde{Z}_k^s$ and then compute $D_{CS}^s$ (using Equation (31));
3: Threshold control: For two predicted target states $PITS_k^i$ and $PITS_k^j$, compute $x_{diff}^k(i,j)$ (using Equation (32)) and $z_{diff}^{s,k}(i,j)$ (using Equation (33)) for each sensor $s \in \mathbb{S}_k^C$. For any $(i,j)$ with both $x_{diff}^k$ and $z_{diff}^{s,k}$ being satisfied simultaneously, modify $D_{CS}^s$ (using Equation (34));
4: Optimization: run the PSO algorithm for solving the objective function (calculated by Equations (35)–(38)), then obtain the optimal sensor subset $\mathbb{S}_k^{A^*}$;
5: Use dynamic clustering strategy to determine CM and CH sensors by their remaining energy, and then complete the measurement data transmission;
6: Rank sensors in $\mathbb{S}_k^{A^*}$ from low CS divergence to high CS divergence, then perform SMC-CBMeMBer filtering with sequential update in this sequence.
7: Update the remaining energy of all sensors in the network and then obtain the candidate sensos set $\mathbb{S}_k^C$ including all live sensors;
**Output:** multi-target density $\pi_{k|k}$ and multi-target state estimate $X_{k|k}$.

---

## 4. Simulations and Performance Analysis

In this section, we present the simulation results in a bearings-only multi-sensor multi-target tracking scenario. Each activated sensor can obtain the measurement data of existing targets by the bearings-only observation model as follows:

$$z_k^s = arctan \frac{p_{x,k} - x^s}{p_{y,k} - y^s} + v_k^s \tag{43}$$

where $z_k^s$ represents the noisy measurement of sensor $s$ at time $k$, $(p_{x,k}, p_{y,k})$ and $(x^s, y^s)$ denote the target position and sensor position, respectively, $v_k^s \sim \mathcal{N}(\cdot; 0, R_k)$ is the measurement noise with $R_k = \sigma_R^2$, and $\sigma_R = (\pi/180)rad$ is the standard deviation of the bearing measurement noise.

Consider that the target moves with a constant velocity (CV) model in this scenario. One target state at time $k$ is denoted by $x_k = \left[ p_{x,k}, \dot{p}_{x,k}, p_{y,k}, \dot{p}_{y,k} \right]^T$, where $(p_{x,k}, p_{y,k})$ and $(\dot{p}_{x,k}, \dot{p}_{y,k})$ are the target position and velocity, respectively. The state transition model is given as

$$x_k = F_k x_{k-1} + u_k \tag{44}$$

where $F_k$ is the transition matrix and $u_k \sim \mathcal{N}(\cdot; 0, Q_k)$ is the process noise with the parameters

$$F_k = \begin{bmatrix} 1 & t & 0 & 0 \\ 0 & 1 & 0 & 0 \\ 0 & 0 & 1 & t \\ 0 & 0 & 0 & 1 \end{bmatrix}, \ Q_k = \sigma_u^2 \begin{bmatrix} t^4/4 & t^3/2 & 0 & 0 \\ t^3/2 & t^2 & 0 & 0 \\ 0 & 0 & t^4/4 & t^3/2 \\ 0 & 0 & t^3/2 & t^2 \end{bmatrix}$$

where $t = 1$s is the sampling period and $\sigma_u = 5 \, \text{m/s}$ is the standard deviation of the process noise.

The birth process is a multi-Bernoulli RFS with density $\pi_\tau = \left\{ (r_\tau, p_\tau^{(i)}) \right\}_{i=1}^4$ where $p_\tau^{(i)}(x) = (x; m_\tau^{(i)}, P_\tau)$. Clutter in the surveillance region follows Poisson distribution with intensity $\lambda_c$. The

parameters used in the SMC-CBMeMBer filter for multi-target tracking filtering are listed in Table 1. More details are given in [15].

**Table 1.** SMC-CBMeMBer filter parameters.

| Parameter | | Value |
|---|---|---|
| Target birth | Existence probability $r_\tau$ | 0.03 |
| | Birth state $m_\tau^{(1)}$ | $[0, 0, 0, 0]^T$ |
| | Birth state $m_\tau^{(2)}$ | $[400, 0, -600, 0]^T$ |
| | Birth state $m_\tau^{(3)}$ | $[-800, 0, -200, 0]^T$ |
| | Birth state $m_\tau^{(4)}$ | $[-200, 0, 800, 0]^T$ |
| | Covariance matrix $P_\tau$ | $diag([10, 10, 10, 10]^T)^2$ |
| Survival probability $p_{s,k}$ | | 0.99 |
| Detection probability $p_{D,k}$ | | 0.98 |
| Clutter intensity $\lambda_c$ | | $0.5 \times 10^{-7} \mathrm{m}^{-2}$ |
| Pruning of hypothesized tracks | Weight threshold $P$ | $10^{-3}$ |
| | Maximum of tracks $T_{max}$ | 100 |
| Number of particles per track | Maximum of particles $L_{max}$ | 1000 |
| | Minimum of particles $L_{min}$ | 300 |

Here we consider that there are 100 sensors randomly distributed over a two-dimensional surveillance region $[-1000 \text{ m}, 1000 \text{ m}]^2$. A maximum of five targets appear in this surveillance region, and the number of targets is time-varying. Target births and deaths occur at various times and locations. The target trajectories and sensor distribution are shown in Figure 5. The start and stop positions for each target are also marked in this figure.

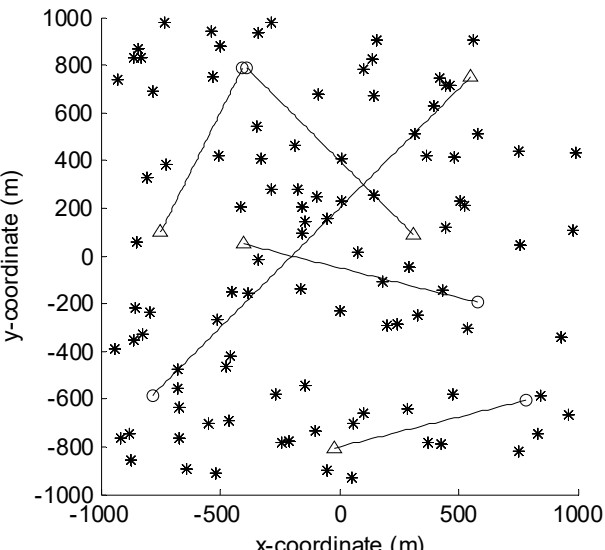

**Figure 5.** Target trajectories and sensor distribution in the x/y plane. Start/stop positions for each target are marked by ∘/Δ, and sensor positions are marked by *.

In order to demonstrate the performance of our proposed solution, all the following solutions are performed for comparison and analysis: the CS divergence center with threshold control proposed in this paper, and sensors with optimal CS divergence and random selection, namely CSC-TC, CS, and RAND. As one of the compared methods, the CS solution can represent the implicit optimal tracking accuracy because it selects sensors with the highest information gain, but it does not consider any energy elements for sensor selection. Therefore, it can be used to verify the validity and significance of our proposed solution in terms of tracking accuracy and energy consumption. The other compared RAND solution is very simple to implement with low complexity, but its tracking accuracy is

often considered inferior because of the randomness of sensor selection. However, it can provide specific evidence in support of sensor management. The above three solutions are implemented over 100 Monte Carlo (MC) trials. At each trial, the same target trajectories and sensor distribution shown in Figure 5 are used but new measurement data are randomly generated.

The number of active sensors selected at each time step is fixed to $N_a = 3$. The two threshold control parameters in CSC-TC are $\varepsilon_z = 0.1$ rad and $\varepsilon_x = 200$ m. Because of the similar scenario in WSNs [69], the PSO parameters are determined as listed in Table 2 to find the optimal CS divergence center. It should be noted that the optimization performance can be improved for better sensor selection results by optimized parameters or other optimization algorithms. Here, it is enough to verify the feasibility and effectiveness of the proposed solution using the parameters shown in Table 2. Figure 6 shows the target position estimate results for a single MC run. From this figure, it can be seen that all three solutions seem able to give acceptable tracking results. They can all identify target births, deaths, and crossings in a cluttered environment. However, the RAND solution has more individuals scattered outside the true track than others. By comparison, CSC-TC has the most accurate estimate fitting to the target trajectories.

**Table 2.** PSO parameters.

| Parameter | Value |
| --- | --- |
| Population size *popsize* | 20 |
| Population initialization *pop* | random generation in surveillance region |
| The velocity of particles $v$ | random generation between $[-10, 10]$ |
| The inertia weight factor $\omega$ | 0.5 |
| The acceleration coefficient $\varphi_1$ | 0.4 |
| The acceleration coefficient $\varphi_2$ | 0.6 |
| Maximum iterations *maxgen* | 30 |

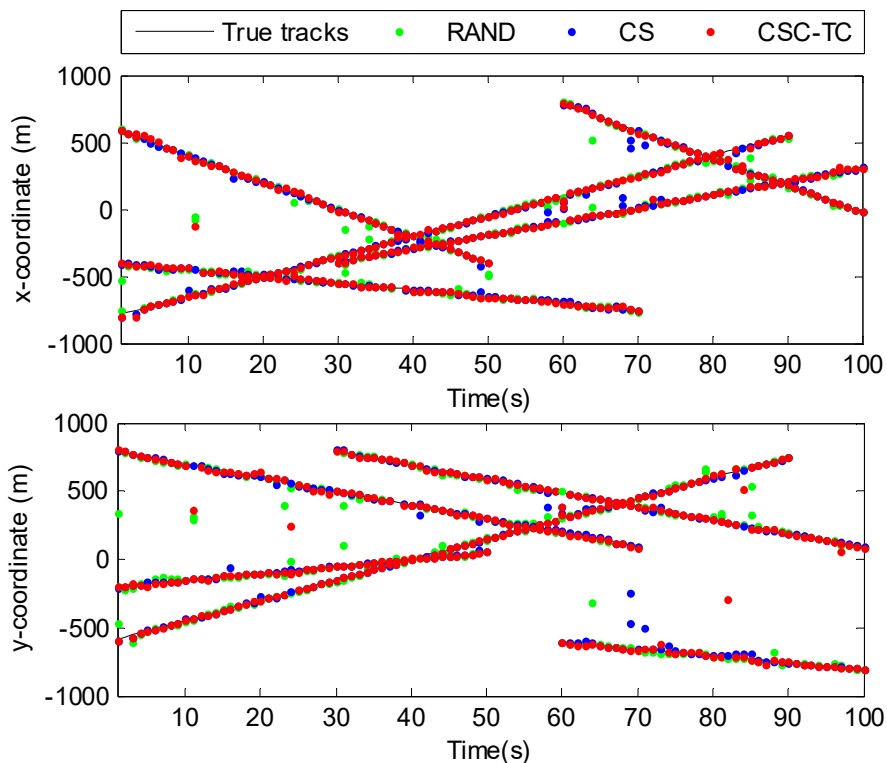

**Figure 6.** True track and target position estimate results.

Figure 7 shows the MC averages of the OSPA distance (with $p = 1$ and $c = 300$) versus time. It can be seen that CSC-TC can achieve almost the same multi-target tracking performance as CS. There are two implicit reasons for the good performance of CSC-TC: First, the sensor subset selected by CSC-TC has a maximum sum of CS divergence as a sensor cluster, which means the sensors can be expected to reach good tracking performance because of their high CS divergence level. Second, the

threshold control in CSC-TC can effectively reduce the effect of the bearing-only sensors with high CS divergence but cannot distinguish different target positions. The RAND solution as evidence of sensor management has the biggest errors because the sensors are selected randomly with an equal probability at each time step. It is unable to determine or control tracking accuracy because of the high randomness. We can also observe that the CS divergence in the CSC-TC and CS solutions does not show an advantage in OSPA distance at the beginning stage of the tracking. It is considered that the initial particles in SMC-CBMeMBer are randomly generated in the surveillance region, and the sensor selection by CS divergence cannot have a very positive effect in tracking accuracy because of the particles scattered away from the true target position. In addition, the sensors selected by clustering in CSC-TC are concentrated near each other. However, there are multiple targets that occur at different positions in the beginning, and they are very far apart. The scattered sensors can perform SMC-CBMeMBer filtering better, especially for bearings-only tracking. As time goes on, the sensor selections by the CS divergence in CSC-TC and CS have more obvious advantages than RAND.

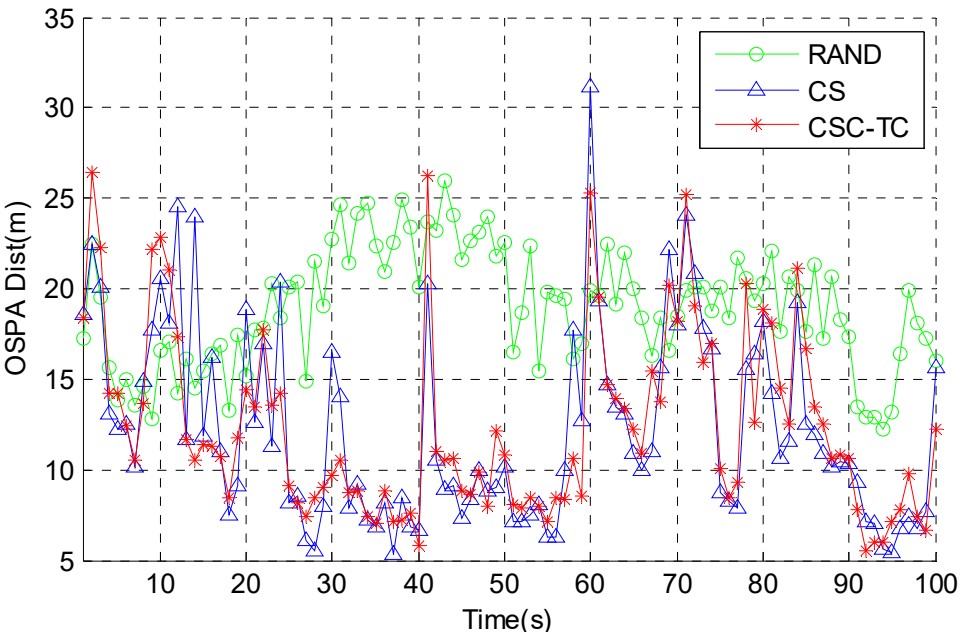

**Figure 7.** OSPA distances for CSC-TC, CS, and RAND.

From the MC averages of the OSPA distance (Figure 7), we can also find that some large error points appear in the CSC-TC and CS solutions. To explain this phenomenon more clearly, some details of sensor selection in CSC-TC and CS are listed in Tables 3 and 4, respectively. The data given in these tables are taken from the target tracking process for a single MC run, which can explain the sudden decreases in tracking accuracy at some times. It can be seen that there are some selected sensors that have little difference between measurements, but the corresponding targets are far apart. For example, at time $k = 9$, 11, 41, 60, and 84 in Table 3 and time $k = 10$, 12, 41, 60, and 71 in Table 4, these sensors and their measurements are shown in bold. They may result in incorrect target state estimates because of the similar bearings-only measurements. The CS solution only calculates the CS divergence for sensor selection but will not consider the bearings-only measurement problem. Although the threshold control method in CSC-TC can reduce the impact to a certain extent compared with the CS solution, some inappropriate sensors are still selected. The possible reasons are as follows: First, the threshold control is implemented based on the predicted ideal measurement and predicted ideal target position, but these are not the same as the real measurement and real target position, respectively, so it cannot avoid selecting these inappropriate sensors completely. Second, the threshold parameter setting has an effect on tracking performance, which is given with experience in this paper. Increasing the threshold $\varepsilon_z$ can exclude more sensors that have the possibility of inaccurate tracking, but otherwise it may be more beneficial for selecting sensors with lower CS divergence, which will reduce overall tracking accuracy. Furthermore, when multiple targets are close together—for example, there are two close even cross-moving targets at time $k = 41$—the threshold control will not be performed because the condition of the predicted ideal target position difference is not satisfied.

**Table 3.** Sensor selection details for CSC-TC.

| Time | Target Position | Sensor Selection | Sensor Position | Measurements * |
|---|---|---|---|---|
| $k = 9$ | T1: (−445, 710)<br>T2: (420, −155)<br>T3: (−665, −465) | S1: 31<br>**S2: 15**<br>S3: 4 | S1: (−673.5399, −637.0967)<br>S2: (−676.6684, −554.8164)<br>S3: (−678.5361, −473.1620) | S1: [0.1799 1.1491 0.0525]<br>S2: [**0.1952** 1.2168 **0.1240**]<br>S3: [0.2155 1.2746 1.0528] |
| $k = 11$ | T1: (−455, 690)<br>T2: (380, −145)<br>T3: (−635, −435) | S1: 81<br>S2: 54<br>**S3: 9** | S1: (−502.9808, 878.2477)<br>S2: (−343.0001, 935.0870)<br>S3: (−530.8344, 754.1318) | S1: [2.8907 2.4469 3.2643]<br>S2: [3.6034 2.5462 3.3676]<br>S3: [**2.3021 2.3828** 3.2264] |
| $k = 41$ | T1: (−605, 390)<br>**T2: (−220, 5)**<br>**T3: (−185, 15)**<br>T4: (−280, 680) | S1: 36<br>**S2: 1**<br>S3: 53 | S1: (−144.9029, 144.6242)<br>S2: (−163.6571, −137.9836)<br>S3: (−156.4971, 95.5383) | S1: [5.1852 3.6387 3.4382 6.047]<br>S2: [5.5811 5.9062 **6.1385 6.1043**]<br>S3: [5.3050 3.7520 3.4959 6.0728] |
| $k = 60$ | T1: (−700, 200)<br>T2: (100, 300)<br>T3: (−90, 490)<br>T4: (780, −605) | **S1: 90**<br>**S2: 89**<br>**S3: 19** | S1: (930.5675, −341.4665)<br>S2: (539.4818, −302.1271)<br>S3: (843.1165, −583.6014) | S1: [5.0168 **5.3709 5.3832** 3.6625]<br>S2: [5.1041 **5.6608 5.6113** 2.4422]<br>S3: [5.1958 **5.5955 5.5741** 4.3671] |
| $k = 84$ | T1: (460, 660)<br>T2: (150, 250)<br>T3: (300, −725) | S1: 58<br>S2: 43<br>**S3: 99** | S1: (419.9635, 743.6434)<br>S2: (446.8321, 717.6404)<br>S3: (464.7387, 712.6752) | S1: [2.6678 3.6083 3.2350]<br>S2: [2.9175 3.6913 3.2450]<br>S3: [**3.2142** 3.7250 **3.2340**] |

\* Measurements: no clutter and no missed detection.

**Table 4.** Sensor selection details for CS.

| Time | Target Position | Sensor Selection | Sensor Position | Measurements * |
|---|---|---|---|---|
| $k = 10$ | T1: (−450, 700)<br>T2: (400, −15)<br>T3: (−650, −450) | **S1: 19**<br>S2: 4<br>**S3: 92** | S1: (843.1165, −583.6014)<br>S2: (−678.5361, −473.1620)<br>S3: (432.6182, −143.2534) | S1: [**5.4974 5.5089** 4.7978]<br>S2: [0.1944 1.2805 0.8721]<br>S3: [5.4751 **4.5264 4.4468**] |
| $k = 12$ | T1: (−460, 680)<br>T2: (360, −140)<br>T3: (−620, −420) | S1: 92<br>**S2: 19**<br>S3: 55 | S1: (432.6182, −143.2534)<br>S2: (843.1165, −583.6014)<br>S3: (−478.4727, −465.9557) | S1: [5.4438 4.7506 4.4633]<br>S2: [**5.4847 5.4496** 4.8183]<br>S3: [6.2710 1.1885 5.0214] |
| $k = 41$ | T1: (−605, 390)<br>**T2: (−220, 5)**<br>**T3: (−185, 15)**<br>T4: (−280, 680) | S1: 80<br>S2: 53<br>**S3: 1** | S1: (−411.8393, 205.5174)<br>S2: (−156.4971, 95.5383)<br>S3: (−163.6571, −137.9836) | S1: [5.4884 2.3698 2.2687 0.2706]<br>S2: [5.3050 3.7520 3.4959 6.0728]<br>S3: [5.5811 5.9062 **6.1385 6.1043**] |
| $k = 60$ | T1: (−700, 200)<br>T2: (100, 300)<br>T3: (−90, 490)<br>T4: (780, −605) | **S1: 18**<br>**S2: 39**<br>**S3: 19** | S1: (10.3578, 408.7489)<br>S2: (144.1419, 257.6447)<br>S3: (843.1165, −583.6014) | S1: [4.4338 **2.4627** 5.3945 **2.4854**]<br>S2: [4.6687 **5.4718 5.4913** 2.5241]<br>S3: [5.1958 **5.5955 5.5741** 4.3671] |
| $k = 71$ | T1: (265, 465)<br>T2: (20, 380)<br>T3: (560, −660) | S1: 66<br>S2: 8<br>**S3: 18** | S1: (313.5009, 511.3323)<br>S2: (426.0601, −790.5447)<br>S3: (10.3578, 408.7489) | S1: [3.9714 4.2779 2.9518]<br>S2: [6.1574 5.9508 0.7933]<br>S3: [1.3465 **2.7801 2.6861**] |

\* Measurements: no clutter and no missed detection.

The MC averages of cardinality estimates for all solutions versus time are also shown in Figure 8. We can see that all solutions are able to achieve a very accurate estimate of the number of targets over the MC simulation. The sparse clutter and occasional detection misses will not have a fatal influence on the tracking performance in most cases.

To evaluate the performance in terms of energy consumption, the communication model for WSNs [70,71] is used as follows:

$$E_t = l_e \cdot \varepsilon_{elec} + l_e \cdot \varepsilon_{amp} \cdot d^4 \tag{45}$$

$$E_r = l_e \cdot \varepsilon_{elec} \tag{46}$$

where $E_t$ represents the energy to transmit $l_e$ bits of data $d$ meters, $E_r$ is the energy to receive $l_e$ bits of data, $\varepsilon_{elec}$ is the energy per bit to run the electronics, and $\varepsilon_{amp}$ is the energy per bit to run the power amplifier. In this paper, $\varepsilon_{elec} = 0.5 \times 10^{-7}$ J/bit, $\varepsilon_{amp} = 1.3 \times 10^{-14}$ J/bit/m$^4$, and $l_e = 500$ bits. The

position of the base station is [0, 0]. In CS and RAND solutions, the selected sensors receive the activated information from the base station and send their bearing measurement data to the base station. In the CSC-TC solution, the CM sensors send their measurements to their CH sensor, and then the CH sensor sends all measurement data to the base station.

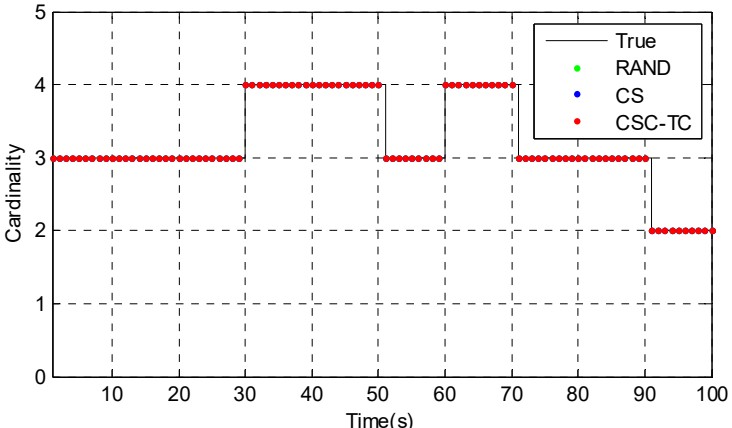

**Figure 8.** Cardinality estimates for CSC-TC, CS, and RAND. (The estimates for all solutions are exactly the same and the points for RAND and CS are overlapped by CSC-TC in the figure).

It is also important to note that the computation is mainly performed by the base station, containing the CS divergence calculation, the PSO operation, and SMC-CBMeMBer filtering within one time step. Generally, the computing and processing power of the base station is considered to be great and continuously improving, so the energy consumption is only considered in the sensors. Meanwhile, the sensor load is not increased by the sensor management solution, which is still just the basic function of data transmission and target observation.

Figures 9 and 10 show the energy consumption results for different sensor management solutions. From Figure 9, it can be easily seen that CSC-TC consumes the least amount of energy, and this is because the sensors selected by CSC-TC can save more energy by close clustering and shorter intra-cluster communication distance. The sensors selected by CS are closer to the targets. According to the target trajectories, the targets in this scenario appear relatively near the boundary and move toward the center of the surveillance region. Therefore, the communication between sensors and the base station in CS consumes less energy than in RAND because of the shorter communication distance. RAND uses more sensors distant from the base station and consumes the most amount of energy.

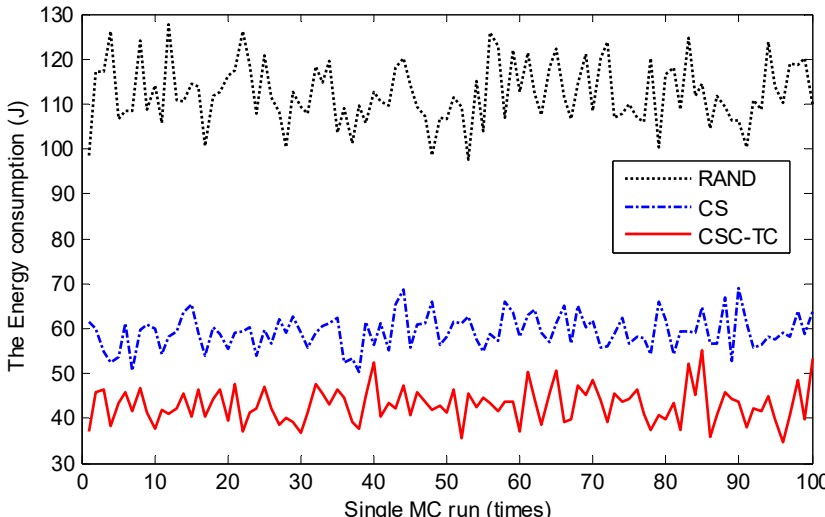

**Figure 9.** The total energy consumption for each single MC run.

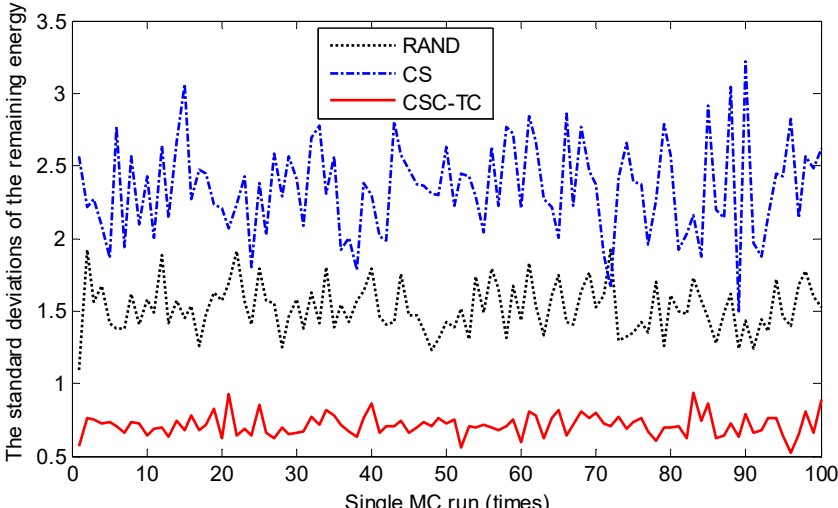

**Figure 10.** The standard deviations of the remaining energy for all sensors in the network at each single MC run.

Moreover, the standard deviations of the remaining energy for all sensors in the network at each single MC run are shown in Figure 10. The CSC-TC solution has a prominent advantage in the energy balance of the network. Its standard deviation of the remaining energy is much lower than those of the other two solutions. This is mainly due to the cluster strategy that uses the sensor with the largest remaining energy as cluster head to take on the communication task with the base station. From Figure 10, we can also see that the CS solution has the biggest standard deviation because it always selects sensors with the highest CS divergence. These sensors are selected repeatedly, and this results in a serious energy imbalance in the entire network.

Table 5 gives the statistical averages of the comprehensive results. The first column gives the average total energy consumption for all MC runs. CSC-TC has lower energy consumption than the other two solutions. This is consistent with the result in Figure 9 and more directly reflects the difference between the solutions. The second column shows the average of the standard deviation of the remaining energy for all MC runs. It can further verify the necessity and the importance of the dynamic clustering strategy in CSC-TC. This strategy can help CSC-TC achieve energy efficiency and energy balancing with good tracking performance. The final column gives the average of the OSPA distance for all the times, which clearly proves that CSC-TC has a tracking accuracy almost as good as CS on the whole, though it does not select sensors with the highest CS divergence.

**Table 5.** Statistical averages of the comprehensive results.

| Solution | Total Energy Consumption (J) | The Standard Deviation of Remaining Energy | OSPA Distance (m) |
|---|---|---|---|
| CSC-TC | 42.9779 | 0.7146 | 12.3957 |
| CS | 59.3069 | 2.3454 | 12.3025 |
| RAND | 112.3411 | 1.5195 | 18.9313 |

In order to further demonstrate the effect of the clustering strategy, the sensor selection results by three different solutions are shown in Figure 11. As we know, the cluster formed by CH and CM sensors at each time step is activated for tracking in CSC-TC. Figure 11a shows all CH and CM sensors, which are represented by red circles and blue circles, respectively. Figure 11b,c show all selected sensors in the CS and RAND solutions, in both cases denoted by blue circles. For CSC-TC and CS, it is easy to find that the selected sensors gather around the area in which the targets occur, and the sensors near the boundary of the surveillance region are relatively rarely selected because no target appears. This more strongly confirms that the energy consumption is unbalanced for the network, which affects the network lifetime. In the RAND solution, all sensors have the same probability of being selected (Figure 11c), but the random selection will result in lower accuracy and more energy consumption.

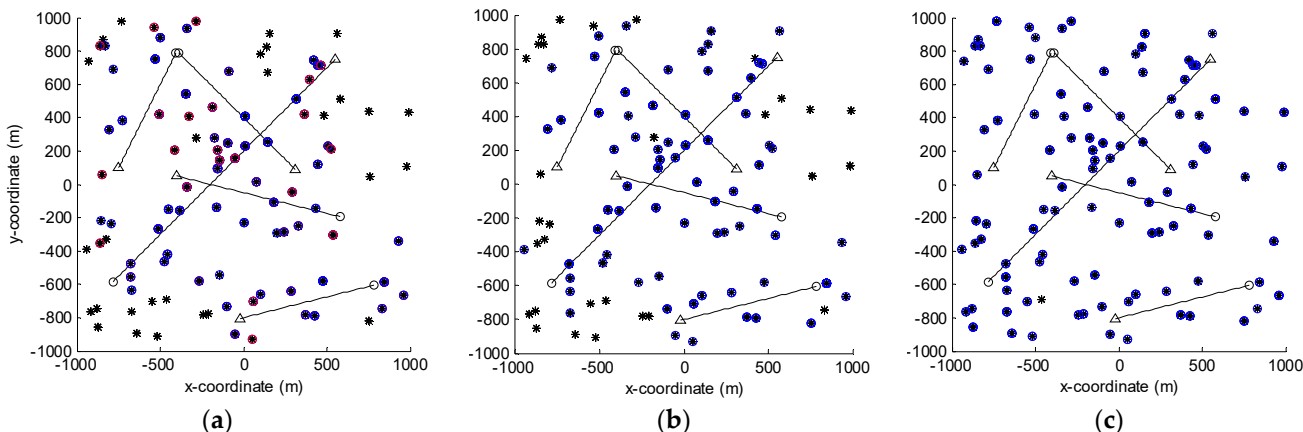

**Figure 11.** The comparison of sensor selection: (**a**) CSC-TC; (**b**) CS; and (**c**) RAND. Start/stop positions for each target are marked by ∘/∆, and sensor positions are marked by *. The CH sensors are marked by red circles, and all other activated sensors are marked by bule circles.

The details of sensor selection results at representative times are listed in Table 6 for easier comparison. From these data, it can be found that the sensors selected by three solutions have significantly different distributions. The clustering strategy in CSC-TC results in the selected sensors being close together, and data collection is completed by intra-cluster transmission, thus prolonging the lifetime of the sensors with less remaining energy. This is very important for CSC-TC to have better performance in terms of energy balance and network lifetime. In the CS solution, the sensors near targets tend to be selected because the CS divergence of these sensors is generally higher than that of other sensors. The selected sensors have no regularity in the RAND solution. Here, note that there is no known information about the target position when making decisions for sensor selection at the initial time, so sensor selection is mainly based on the possible target birth at time $k = 1$.

**Table 6.** The details of sensor selection comparison at representative times.

| Time | Target Position | Sensor Selection * | Sensor Position |
|---|---|---|---|
| $k = 1$ | T1: (−405, 790) <br> T2: (580, −195) <br> T3: (−785, −585) | CSC-TC: {**29**,76,69} <br> CS: {53,47,69} <br> RAND: {41,38,57} | CSC-TC: (181.4109, −104.5588), (−49.4442, 155.5188), (74.0502, 13.6267) <br> CS: (−156.4971, 95.5383), (9.8250, 232.5366), (74.0502, 13.6267) <br> RAND: (−52.9732, −896.1277), (−854.8679, −216.3499), (−874.8099, −854.2625) |
| $k = 20$ | T1: (−500, 600) <br> T2: (200, −100) <br> T3: (−500, −300) | CSC-TC: {**16**,20,40} <br> CS: {19,40,29} <br> RAND: {26,44,36} | CSC-TC: (−380.7319, −154.5384), (−452.3069, −152.1786), (−512.0281, −265.9178) <br> CS: (843.1165, −583.6014), (−512.0281, −265.9178), (181.4109, −104.5588) <br> RAND: (473.6086, −580.2009), (−175.6236, 278.7173), (−144.9029, 144.6242) |
| $k = 40$ | T1: (−600, 400) <br> T2: (−200, 0) <br> T3: (−200, 0) <br> T4: (−290, 690) | CSC-TC: {**78**,95,37} <br> CS: {9,37,19} <br><br> RAND: {19,7,93} | CSC-TC: (−727.9327, 382.1973), (−349.6038, 543.7587), (−506.9295, 421.9507) <br> CS: (−530.8344, 754.1318), (−506.9295, 421.9507), (843.1165, −583.6014) <br><br> RAND: (843.1165, −583.6014), (159.0102, 904.9361), (−95.4791, 248.1324) |
| $k = 60$ | T1: (−700, 200) <br> T2: (100, 300) <br> T3: (−90, 490) <br> T4: (780, −605) | CSC-TC: {**90**,89,19} <br> CS: {18,39,19} <br><br> RAND: {88,46,68} | CSC-TC: (930.5675, −341.4665), (539.4818, −302.1271), (843.1165, −583.6014) <br> CS: (10.3578, 408.7489), (144.1419, 257.6447), (843.1165, −583.6014) <br><br> RAND: (5.0484, −230.3810), (−917.7173, −766.5895), (51.3831, −929.3055) |
| $k = 80$ | T1: (400, 600) <br> T2: (110, 290) <br> T3: (380, −705) | CSC-TC: {66,18,**39**} <br> CS: {28,48,19} <br> RAND: {31,90,56} | CSC-TC: (313.5009, 511.3323), (10.3578, 408.7489), (144.1419, 257.6447) <br> CS: (282.6695, −642.0494), (369.0952, −782.5874), (843.1165, −583.6014) <br> RAND: (−673.5399, −637.0967), (930.5675, −341.4665), (748.6628, 440.7059) |
| $k = 100$ | T1: (310, 90) <br> T2: (−20, −805) | CSC-TC: {**96**,41,73} <br> CS: {73,52,19} <br> RAND: {75,13,73} | CSC-TC: (56.6925, −704.5359), (−52.9732, −896.1277), (−100.4117, −735.8783) <br> CS: (−100.4117, −735.8783), (291.4503, −43.4087), (843.1165, −583.6014) <br> RAND: (328.8144, −245.8045), (−283.2894, 975.5527), (−100.4117, −735.8783) |

\* Sensor selection: the selected sensors are listed in sequential update order. The sensor in bold is the CH sensor for CSC-TC.

Additionally, we explore the network performance in terms of network lifetime. Lifetime is usually evaluated based on the time when the first sensor dies. The different solutions for the same trajectories as in Figure 5 are executed for 100 rounds by continuous running. The sensors will die

with time because of exhausted energy. As we observed earlier, the sensors are selected with uneven distribution due to the different locations of sensors.

Table 7 lists the time when the first sensor dies in CSC-TC, CS, and RAND. It can be seen that the first sensor death time is the earliest in CS and latest in CSC-TC. As we know, the sensors with high CS divergence will be activated often, and this will accelerate the death of these sensors. Therefore, the first sensor dies at the 119th round, which is earlier than in the other two solutions. In Figure 12, the number of live sensors versus the round of running algorithms is shown for further demonstration. It can be clearly seen that the number of live sensors decreases with time, and CSC-TC always has more live sensors than the other two solutions. For most of the times, the number of live sensors for RAND is the lowest. This is because RAND consumes the most amount of energy for communication, which is proved by the energy consumption data in Table 5. It is clear that CSC-TC has the longest network lifetime, which is effectively prolonged by the clustering strategy, in contrast to CS.

**Table 7.** The comparison of the first sensor death time.

| Solution | CSC-TC | CS | RAND |
|---|---|---|---|
| First senor death time (s) | 1533 | 119 | 319 |

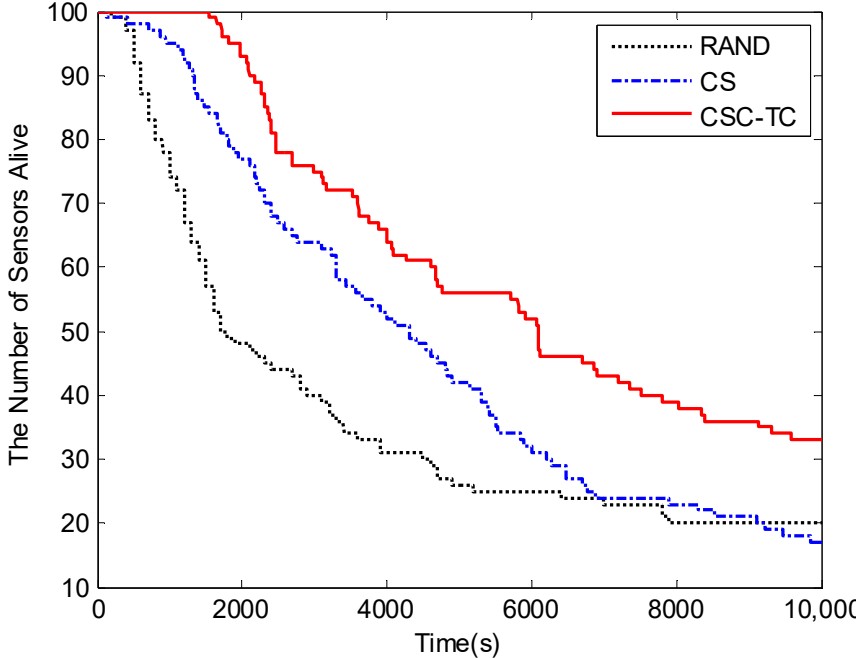

**Figure 12.** Network lifetime.

## 5. Conclusions

This paper studied the sensor management problem for bearings-only multi-target tracking in WSNs. This study has identified the importance of sensor selection for tracking performance and energy efficiency. It has also shown that the bearings-only measurement will have a non-ignorable impact on the target state estimate by sequential update. The other important considerations in this paper are the energy efficiency and energy balancing of the network. The proposed solution adopts a threshold control method to further ensure bearings-only tracking accuracy, defines an information center to select sensors by their information gain, constructs a new objective function using swarm intelligence optimization, and balances the network energy consumption by a dynamic clustering strategy. The simulation results prove the good performance of the proposed solution. Nevertheless, there is ample scope for future research. The data fusion approaches for bearings-only measurements still remain to be studied. We also need to deeply consider the equilibrium problem between tracking accuracy and energy consumption. Finally, the effect of dense clutter on bearings-only tracking should be considered for filtering.

**Author Contributions:** Conceptualization, X.J. and T.M.; methodology, X.J. and J.J.; software, X.J. and Y.J.; investigation, X.J. and T.M.; writing—original draft preparation, X.J.; writing—review and editing, X.J., T.M., J.J. and Y.J.; funding acquisition, X.J. All authors have read and agreed to the published version of the manuscript.

**Funding:** This research was funded by the National Natural Science Foundation of China under grant 61701295.

**Data Availability Statement:** Not applicable.

**Conflicts of Interest:** The authors declare no conflict of interest.

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
