# Peer review of "Sensor Management with Dynamic Clustering for Bearings-Only Multi-Target Tracking via Swarm Intelligence Optimization"

_electronics, doi:10.3390/electronics12163397_

Round 1
Reviewer 1 Report
In this paper, a sensor management method is proposed to solve the problem of high nonlinearity and poor observability in bearings-only sensor networks, while considering the energy efficiency and energy balancing. This paper introduces the basic knowledge of multi-target tracking, and proposes a sensor management solution, including optimal sensor subset selection, bearings-only measurement threshold control, objective function construction, swarm intelligence optimization and dynamic clustering strategy, to achieve good tracking accuracy while considering energy efficiency and balancing. Simulations verify the effectiveness of the proposed method. This paper is basically well motivated and technically sound. It can be considered for publication after some revision and improvement.
Here are some suggestions:
1. In the context of distributed RFS filtering by using WSNs, the (arithmetic) average density fusion has recently received considerable attention and demonstrated great performance in many scenarios with the use of either PHD/CPHD filter or the Bernoulli/MB filters. The authors should not overlook these approaches in the literature study and even in the simulation comparison (if possible).
2. The authors should highlight that the filter used in the paper is the MB filter in the Abstract.
3. Formula format problems need to be noted. For example, in Equations (12) and (14), the length of the division sign is not the same; the length of parentheses in different formulas is not uniform.
4. More objective functions and optimization methods can be compared to better verify the performance of the proposed method.
5. The computational complexity of the proposed method is an issue worthy of attention and analysis.
The work needs to be further polished. Language expression needs to be optimized, such as the description of multi-target density update in the case of three active sensors in lines 368 to 375 is rather lengthy.
Reviewer 2 Report
In the presented paper, authors propose threshold control method to reduce the impact on tracking accuracy when using bearings-only measurements and use PSO to construct a new objective function for efficiently finding the optimal sensor subset in WSN.
The paper is overall well written and clear.
However, I have following comments to the authors:
1. The authors should provide the information on how did they determined the parameters used in PSO algorithm? Did they conduct tests with different values?
2. When evaluating the performance on energy consumption, authors use two equations, namely 45 and 46. Are these some well-known equations from the existing literature, or the authors invented them. If they are already used in previous papers, they should be referenced., otherwise, the authors should explain why they constructed them this way.
3. The Figure 11. provides interesting insight into the behavior of different methods. However, I would recommend adding data about the order of the activated clusters/sensors, if possible. It would be far more informative if the reader could see which sensor was selected while the target was moving at some representative time moments. For example, if we compare only selected sensors in F 11(a) and F 11(b) (blue dots) there is little difference. But, with adding recommended data, we believe that the difference will be clearer.
Reviewer 3 Report
The manuscript contains novel elements. However, it presents some aspects that need to be solved before reconsideration.
The authors should explicitly mention the significant contributions of the manuscript. The novelty of the paper is not highlighted.
In the introduction make a small survey of swarm intelligence methods including approaches such as:
PSO (Particle swarm optimization) Kennedy, J., & Eberhart, R. (1995, November). Particle swarm optimization. In Proceedings of ICNN'95-international conference on neural networks (Vol. 4, pp. 1942-1948). IEEE.
SSO (Social Splider Optimization) Cuevas, E., Cienfuegos, M., Zaldívar, D., & Pérez-Cisneros, M. (2013). A swarm optimization algorithm inspired in the behavior of the social-spider. Expert Systems with Applications, 40(16), 6374-6384.
Firefly Yang, X. S., & Slowik, A. (2020). Firefly algorithm. In Swarm intelligence algorithms (pp. 163-174). CRC Press.
The advantages and limitations of the compared schemes in relationship with similar schemes is not clear.
Please revise the structure of the paper. It is recommendable to add in each section a couple of sentences that explain which is the purpose of the section. With this organization, the reader can clearly understand the sequence of the paper.
Round 2
Reviewer 3 Report
Alll my comments have been addresed